# The Evolutionary Landscape of SARS-CoV-2 Variant B.1.1.519 and Its Clinical Impact in Mexico City

**DOI:** 10.3390/v13112182

**Published:** 2021-10-29

**Authors:** Alberto Cedro-Tanda, Laura Gómez-Romero, Nicolás Alcaraz, Guillermo de Anda-Jauregui, Fernando Peñaloza, Bernardo Moreno, Marco A. Escobar-Arrazola, Oscar A. Ramirez-Vega, Paulina Munguia-Garza, Francisco Garcia-Cardenas, Mireya Cisneros-Villanueva, Jose L. Moreno-Camacho, Jorge Rodriguez-Gallegos, Marco A. Luna-Ruiz Esparza, Miguel A. Fernández Rojas, Alfredo Mendoza-Vargas, Juan Pablo Reyes-Grajeda, Abraham Campos-Romero, Ofelia Angulo, Rosaura Ruiz, Claudia Sheinbaum-Pardo, José Sifuentes-Osornio, David Kershenobich, Alfredo Hidalgo-Miranda, Luis A. Herrera

**Affiliations:** 1Instituto Nacional de Medicina Genómica, INMEGEN, Periferico Sur 4809, Arenal Tepepan, Tlalpan, Mexico City 14610, Mexico; acedro@inmegen.gob.mx (A.C.-T.); lgomez@inmegen.gob.mx (L.G.-R.); nicoam@gmail.com (N.A.); gdeanda@inmegen.edu.mx (G.d.A.-J.); fpenaloza@inmegen.gob.mx (F.P.); bmoreno@inmegen.gob.mx (B.M.); franciscojavgc@gmail.com (F.G.-C.); mcisneros@inmegen.gob.mx (M.C.-V.); amendoza@inmegen.gob.mx (A.M.-V.); jreyes@inmegen.gob.mx (J.P.R.-G.); 2Cátedras CONACYT para Jóvenes Investigadores, CONACYT, Av. de los Insurgentes Sur 1582, Crédito Constructor, Benito Juárez, Mexico City 03940, Mexico; 3Unidad de Investigación Biomédica en Cáncer, Instituto Nacional de Cancerología-Instituto de Investigaciones Biomédicas, UNAM, Av. San Fernando 22, Belisario Domínguez Secc 16, Tlalpan, Mexico City 14080, Mexico; marcoarmandoea@gmail.com (M.A.E.-A.); oscararv5@ciencias.unam.mx (O.A.R.-V.); pmung96@gmail.com (P.M.-G.); 4Clinical Laboratory Division, Salud Digna, Culiacan, Sinaloa 80000, Mexico; jose.moreno@salud-digna.org (J.L.M.-C.); jorge.rodriguez@salud-digna.org (J.R.-G.); 5Molecular Biology Laboratory, National Reference Center, Salud Digna, Tlalnepantla de Baz, Estado de Mexico 54075, Mexico; 6Innovation and Research Department, Salud Digna, Culiacan, Sinaloa 80000, Mexico; marco.luna@salud-digna.org (M.A.L.-R.E.); miguel.fernandez@salud-digna.org (M.A.F.R.); abraham.campos@salud-digna.org (A.C.-R.); 7Secretaría de Educación, Ciencia, Tecnología e Innovacion, Av Chapultepec 49, Colonia Centro, Cuauhtémoc, Mexico City 06010, Mexico; ofelia.angulo@sectei.cdmx.gob.mx (O.A.); rosaura.ruiz@educacion.cdmx.gob.mx (R.R.); 8Gobierno de la Ciudad de México, Antiguo Palacio del Ayuntamiento, Avenida Plaza de la Constitución 2, Colonia Centro, Mexico City 06010, Mexico; csheinbaump@ii.unam.mx; 9Instituto Nacional de Ciencias Médicas y Nutrición Salvador Zubirán, Vasco de Quiroga 15, Belisario Domínguez Secc 16, Tlalpan, Mexico City 14080, Mexico; jose.sifuentes@incmnsz.mx (J.S.-O.); david.kershenobichs@incmnsz.mx (D.K.)

**Keywords:** SARS-CoV-2, B.1.1.519 variant, effective reproduction number, phylogenetic analysis, haplotype analysis, clinical impact, significant adjusted odds ratio, COVID-19 hospitalization, COVID-19 critical outcome

## Abstract

The SARS-CoV-2 pandemic is one of the most concerning health problems around the globe. We reported the emergence of SARS-CoV-2 variant B.1.1.519 in Mexico City. We reported the effective reproduction number (Rt) of B.1.1.519 and presented evidence of its geographical origin based on phylogenetic analysis. We also studied its evolution via haplotype analysis and identified the most recurrent haplotypes. Finally, we studied the clinical impact of B.1.1.519. The B.1.1.519 variant was predominant between November 2020 and May 2021, reaching 90% of all cases sequenced in February 2021. It is characterized by three amino acid changes in the spike protein: T478K, P681H, and T732A. Its Rt varies between 0.5 and 2.9. Its geographical origin remain to be investigated. Patients infected with variant B.1.1.519 showed a highly significant adjusted odds ratio (aOR) increase of 1.85 over non-B.1.1.519 patients for developing a severe/critical outcome (*p* = 0.000296, 1.33–2.6 95% CI) and a 2.35-fold increase for hospitalization (*p* = 0.005, 1.32–4.34 95% CI). The continuous monitoring of this and other variants will be required to control the ongoing pandemic as it evolves.

## 1. Introduction

Severe acute respiratory syndrome coronavirus 2 (SARS-CoV-2) is the etiological cause of Coronavirus Disease 19 (COVID-19), and it has caused the largest and most severe pandemic of this century [1].

Evaluation of the correlation between comorbidities and worse prognoses of COVID-19 disease has shown that hypertension, obesity, and diabetes are the three most prevalent comorbidities. Moreover, cancer, chronic kidney diseases, diabetes mellitus, and hypertension have been associated with mortality in COVID-19 patients [2,3].

SARS-CoV-2 uses the spike protein to infect a host cell by binding to the angiotensin converting enzyme 2 receptor (ACE2) [4]. In addition, the internalization of the virus requires proteolytic activation via the transmembrane protease serine 2 (TMPRSS); it could be activated and regulated by epigenetic modulators [5]. SARS-CoV-2 has a full-length genomic RNA with 29,903 nucleotides, and its sequencing is possible using NGS technologies [6].

Since December 2019, scientists around the world have generated 2.32 M whole-genome sequences of SARS-CoV-2 that have been made publicly available in the Global Initiative on Sharing all Influenza Data (GISAID) initiative database [7]. This massive genome sequencing effort has had an impact on public health and the handling of the pandemic, since it has allowed the design and updating of molecular tests for viral detection [8,9] and guided the design of vaccines and antiviral treatments [10,11]. Moreover, it has enabled the study of viral evolution, with in-depth investigation into the emergence and pursuit of variants of concern (VOCs), such as Alpha, Beta, Gamma, and Delta, and variants of interest (VOIs), such as Eta, Lambda, Iota and Kappa [12].

Monitoring the emergence of new variants of SARS-CoV-2 is a worldwide priority, as alterations such as amino acid substitutions in the viral genome could be related to alterations in biological properties, such as the ligand-like affinity receptor, the neutralization efficiency resulting from naturally acquired immunity or vaccination [13,14,15] or the transmission capacity [16]—as well as the impact on the clinical presentation of the disease [17].

In Mexico, SARS-CoV-2 variants have been monitored since March 2020. During the third epidemic peak between February and March 2021, we observed an increase in variant B.1.1.519, which possesses three substitutions in the spike protein (T478K, P681H, and T732A). This study reported the emergence and spread of the new B.1.1.519 variant in Mexico City and its evolution, transmissibility and association with relevant clinical traits.

## 2. Methods

### 2.1. Sample Preparation

#### 2.1.1. Participants

Nasopharyngeal swabs (NPSs) were collected from 1835 patients for SARS-CoV-2 detection. The study was approved by the ethics and research committees of the Instituto Nacional de Medicina Genómica (CEI/1479/20 and CEI 2020/21).

#### 2.1.2. Sample Collection

NPSs were collected by a trained clinician with a flexible nylon swab that was inserted into the patient’s nostrils to reach the posterior nasopharynx. It was left in place for several seconds and slowly removed while rotating. The swab was then placed in 3 mL of sterile viral transport medium. Swabs from both nostrils were deposited in a single viral transport tube, taken to a clinical laboratory and processed immediately.

#### 2.1.3. SARS-CoV-2 RNA Extraction

Total nucleic acid was extracted from 300 μL of viral transport medium from the NPSs or 300 μL of whole saliva using the MagMAX Viral/Pathogen Nucleic Acid Isolation Kit (Thermo Fisher Scientific, Waltham, MA, USA) and eluted into 75 μL of elution buffer.

#### 2.1.4. RT-qPCR

For SARS-CoV-2 RNA detection, 5 μL of RNA template was tested using the US CDC real-time RT-qPCR primer/probe sets for 2019-nCoV_N1 and 2019-nCoV_N2 and human RNase *p* (RP) as an extraction control. Samples were classified as positive for SARS-CoV-2 when both the N1 and N2 primer/probe sets were detected with a Ct value lower than 40 (Centers for Disease Control and Prevention, Atlanta, GA, USA, 2020). If only one of these genes was detected, the sample was labeled inconclusive. Additionally, RT-PCR was performed using the DA-930-Detection Kit for 2019 Novel Coronavirus (2019-nCoV) RNA (PCR-Fluorescence Probing) (DaAn Gene Co., Ltd. Of Sun Yat-sen University, Guangzhou, China) following the manufacturer’s instructions. All tests were run on Thermo Fisher ABI QuantStudio 5 or QuantStudio 7 real-time thermal cyclers (Thermo Fisher Scientific, Waltham, MA, USA). Samples were selected for inclusion in this study based on viral Ct < 30.

### 2.2. Sequencing

#### 2.2.1. Oxford Nanopore-Sequencing

We performed PCR tiling of the COVID-19 virus, version PTC_9096_v109_revF_06Feb2020. For nanopore amplicon sequencing of SARS-CoV-2, the ARTIC v3 amplification products of each sample were mixed and purified using Agencourt AMPure XP beads (Beckman Coulter, Pasadena, CA, USA) at a 1:1 ratio and finally diluted in 30 μL of water. One microliter of purified DNA amplicons was used for quantification by Qubit™ dsDNA HS Assay Kit (Thermo Fisher Scientific, Waltham, MA, USA). Sequencing library preparation consisted of two steps: native barcode ligation and sequencing adapter ligation. Native barcoding of amplicons was performed in a 20 μL reaction volume (1.5 μL DNA amplicons, 5 ng, 5.5 μL nuclease-free water, 2.5 of Native Barcode EXP-NBD104 and EXP-NBD114 (Oxford Nanopore Technologies, Oxford, Oxfordshire, UK), 10 μL NEBNext Ultra II Ligation Master Mix (New England Biolabs, Ipswich, MA, USA), 0.5 μL of NEBNext Ligation Enhancer (New England Biolabs, Ipswich, MA, USA)) for 20 min at 20 °C and 10 min at 65 °C. The sequencing adapter was ligated in a 50 μL reaction, with 50 ng of 24 barcoded amplicon pools, 10 μL of 5× NEBNext Quick Ligation Reaction Buffer, 5 μL AMII adapter mix, and 5 μL Quick T4 DNA Ligase (New England Biolabs, Ipswich, MA, USA), using an SQK-LSK109 kit (Oxford Nanopore Technologies, Oxford, Oxfordshire, UK). The ligation reaction was performed at room temperature for 20 min. The library was purified using AMPure XP beads and quantified using Qubit™ dsDNA HS Assay Kit. Sequencing was performed on the MinION platform (Oxford Nanopore Technologies, Oxford, Oxfordshire, UK), and the final library (15 ng) was loaded onto the flow cell R.9 according to the manufacturer’s instructions. ONT MinKNOW software (Oxford Nanopore Technologies, Oxford, Oxfordshire, UK) was used to collect raw sequencing data.

#### 2.2.2. Oxford-Nanopore Raw Data Processing and Sequencing Data Quality Assessment

Basecalling and barcode demultiplexing were performed with Guppy (v.4.4.1). Reads were processed according to the ARTIC Network protocols for COVID-19 [18] using a nextflow pipeline (https://github.com/connor-lab/ncov2019-artic-nf, accessed on 1 November 2020). Briefly, for each sample, raw reads were mapped to the Wuhan reference sequence MN908947.3 using primer scheme V3 and Minimap (v.2.17). Post-alignment processing consisted of assigning reads to their derived amplicon and read group based on the primer pool, removing primer sequences, normalizing/reducing the number of read alignments to each amplicon and removing reads with imperfect pairings. Variant calling was performed with medaka (v.1.0.3) on the filtered and trimmed bam files. A final consensus FASTA file was generated by first marking positions not covered by at least 20 reads from either group as low coverage and building a pre-consensus FASTA with BCFtools consensus, which was subsequently aligned against the reference sequence using muscle (v.3.8.1551).

#### 2.2.3. Illumina Sequencing

The libraries were prepared using the Illumina COVID-seq protocol following the manufacturer’s instructions. First-strand synthesis was carried out on RNA samples. The synthesized cDNA was amplified using ARTIC primers V3 for multiplex PCR, generating 98 amplicons across the SARS-CoV-2 genome. The PCR-amplified product was tagmented and adapted using IDT for Illumina Nextera UD Indices Set A, B, C, D (384 indices) (Illumina, San Diego, CA, USA). Dual-indexed single-end sequencing with a 36 bp read length was carried out on the NextSeq 550 platform) (Illumina, San Diego, CA, USA).

#### 2.2.4. Illumina Raw Data Processing and Sequencing Data Quality Assessment

The raw data were processed using DRAGEN Lineage v3.3 with standard parameters (Illumina, San Diego, CA, USA). Further samples with SARS-CoV-2 and at least 90 targets detected were processed for lineage designation.

### 2.3. Genomic Data Collection

Most B.1.1.519 sequences were generated at the Instituto Nacional de Medicina Genómica (INMEGEN) by the abovementioned protocol (*n* = 1710). For completeness, we also downloaded (from GISAID) all sequences from Mexico City with their associated metadata (collection date < 31 May 2021, *n* = 906, not sequenced by INMEGEN). When high-quality sequences were required, we filtered by sequences with at most 1% *n* and less than 0.05% singletons (high coverage) (*n* = 1879). Only INMEGEN samples had associated clinical information, geographical information to the municipality level and a SISVER ID required to retrieve associated information from the federal database. The sample size and metadata used in each analysis is described in Appendix A.

### 2.4. Effective Reproduction Number Estimation for Variants B.1.1.222 and B.1.1.519

We grouped all sequenced samples based on the epidemiological week as the date of sample collection. We then calculated the percentage of samples for the variants of interest B.1.1.222 and B.1.1.519 and the percentage of samples that belonged to the ensemble of other variants. Using these percentages, we extrapolated the total number of confirmed cases using the federal database for residents of Mexico City treated in medical units within Mexico City. With this, we calculated an incidence time series for both variables of interest and the ensemble of other variants.

Using this percentage, we considered all confirmed cases in the federal database for residents of Mexico City treated in medical units within Mexico City. We assumed that these samples were divided in the same percentages as the ones observed in the sequenced samples for a given epidemiological week. Then, we calculated an incidence time series for both variables of interest and the ensemble of other variants.

We estimated the effective reproduction number (Rt) using the parametric method of Cori et al., 2013 [19] and the parameters reported for the SARS-CoV-2 serial interval by Nishiura et al., 2020 [20]. We restricted this analysis to the period beginning with epidemic week 2020–46, corresponding to the first detections of variant B.1.1.519.

### 2.5. Haplotype Analysis for Variant B.1.1.519

Only high-quality sequences were considered. SARS-CoV-2 reference genome NC_045512.2 was downloaded from NCBI. SNVs and indels per SARS-CoV-2 sequence were obtained with nucmer [21]. Nucmer was executed with the following parameters: map each position of each query to its best hit in the reference, map each position of each reference to its best hit in the query and exclude alignments with ambiguous mapping. Variable positions in any SARS-CoV-2 sequence were obtained. Only variable positions observed in at least 5 genomes were further considered. Each SARS-CoV-2 sequence was translated into a compressed representation in which only the genotype of the list of variable positions was included. A unique combination of alleles, e.g., a unique compressed representation, was considered a haplotype. Haplotypes were used to infer a haplotype network using the haploNet function from the Population and Evolutionary Genetics Analysis System package (pegas) [22]. Briefly, genetic distances (Hamming distance) between all pairwise combinations of haplotypes were calculated using the dist.dna function of the Analyses of Phylogenetics and Evolution package (ape) [23]; from this distance matrix, the minimum spanning tree and the median-joining network were computed using pegas [24].

### 2.6. Phylogenetic Analysis

The sequences were aligned with MAFFT (version 7.475) using the FFT-NS-2 algorithm [25,26]. A maximum-likelihood phylogeny was calculated with FastTree (version 2.1.11) and compiled with the double precision tag using a generalized time-reversible model (GTR) [27,28] The resulting tree was visualized using the Interactive Tree Of Life (iTOL) [29].

### 2.7. Clinical Data Collection

To gather and correlate clinical data from our patients, we used the National Epidemiologic Surveillance System for Viral Respiratory Diseases (SISVER). This system gathers information, including personal identification data, contact information, comorbidities, date of diagnosis, symptoms, progression and outcome of all the COVID-19 cases reported in Mexico. After downloading the data collected in this system, we verified and complemented this information with our own variants of interest by applying a telephone survey.

Verbal consent and identification were the first steps when calling each subject included in the final analysis. Questions on our survey covered comorbidities (diabetes, hypertension, cardiovascular diseases, chronic renal failure, COPD, asthma, HIV, cancer, obesity, smoking, pregnancy status and immunosuppression), date of symptom onset, sampling date, COVID-19 symptoms (fever, NSAID-resistant fever, cough, dyspnea, chest pain, oxygen saturation, headache, myalgias, arthralgias, odynophagia, anosmia, ageusia, diarrhea, vomiting, rhinorrhea, polypnea, cyanosis, conjunctivitis and abdominal pain), disease progression (ambulatory or hospitalized). In cases of hospitalized patients, we asked the length of hospital stay, treatment (need for supplementary oxygen or intubation) and outcome (alive, dead or under treatment). For underage or deceased patients, we tried to reach a close relative who was taking care of the individual and could answer all the questions with certainty.

### 2.8. Statistical Analysis

Binary multivariate logistic regression models were fitted to predict the association of symptoms with variant B.1.1.519, as well as the association of hospitalization with the variant. An ordinal multivariate logistic regression model was fitted to predict the association of disease severity with the variant. The severity score was coded as 0 for asymptomatic or mild symptoms, 1 for severe symptoms and 2 for death. Individuals classified with severe disease were those who presented with at least one of the following: dyspnea, polypnea, cyanosis, requiring supplemental oxygen or intubation. All models were adjusted for covariates (age and sex) and comorbidities (immunosuppression, heart disease or hypertension, diabetes, obesity, asthma or smoking).

## 3. Results

### 3.1. Identification of Variant B.1.1.519 in Mexico City

On 3, November, 2020, the first patient carrying variant B.1.1.519 was detected in Mexico City, representing the second case recorded worldwide. The frequency of the B.1.1.519 variant began to increase in Mexico City, from 16% (17/106) to a peak in February 2021 of 90% (496/552). In March 2021, its frequency began to decrease, and in May 2021, it had dropped to 51% (Figure 1A).

Variant B.1.1.519 represented 74.3% of the sequences generated in Mexico City (2296/3092) from November 2020 to May 2021 and was distributed evenly across all of Mexico City (Figure 1B). B.1.1.519 was detected in 31 countries, predominantly in Mexico at (55%, 6041/10,922), followed by the USA (2.2%, 11,937/548,492), Canada (0.87%, 456/52,409) and Germany (0.14%, 192/130,634) by May 2021.

According to the phylogenetic analysis, this variant is grouped in an independent clade derived from the clade 20B NextClade classification (Figure 1C). The B.1.1.159 variant is characterized by 9 mutations (C203T, C222T, C3140T, C10954T, A11117G, C12789T, C21306T, C22995A, and C23604A), four ORF1a substitutions (P959S, T3255I, I3618V, and T4175I) and three spike substitutions (T478K, P681H, and T732A) (Figure 1C). The diversity along the SARS-CoV-2 genome for variant B.1.1.159 is presented in Figure 1D.

### 3.2. Rt: Effective Reproduction Number

We studied the effective reproduction number, defined as the average number of secondary cases per primary case at a given calendar time, to characterize the transmissibility of the B.1.1.519 variant. Matching the rapid increase in detection of variant B.1.1.519, we observed an increase in Rt for variant B.1.1.519 during the month of December 2020 up to a value of 2.9 in the second week of December, before stabilizing between 0.5 and 1 in the following months.

The second most frequent variant in Mexico City was B.1.1.222. All remaining variants had small frequencies and were considered one joint group. Variant B.1.1.222 reached a maximum Rt of 1.93 during the second week of December. Its estimated Rt values fluctuated strongly in the following months, which could be influenced by the small number of cases of this variant. In comparison, the estimated Rt for the ensemble of other variants has fluctuated steadily since the winter, with increases in the first week of January 2021, fourth week of February and second week of March, before stabilizing (Figure 2A,B) or disappearing (Figure 1A).

We compared the number of comorbidities, survival status, hospitalization status, age distribution and geographical distribution between the samples sequenced at INMEGEN and all other samples at the federal database. We found no large differences for either number of comorbidities, survival status, hospitalization status, or age distribution between the different groups (Appendix A). We also compared the percentage of samples per municipality and we observed significant differences for such municipalities, largely due to the fact that the samples analyzed at INMEGEN tended to belong to hospitals, clinical centers or clinical laboratories located near to the INMEGEN (Appendix A). Moreover, as geographical location was the only biased variable, the samples sequenced at INMEGEN represent a suitable representation of the Mexico city population.

### 3.3. Genomic Findings

#### 3.3.1. Phylogenetic Analysis

We calculated a maximum-likelihood phylogeny, including all SARS-CoV-2 genomes of interest (Methods), to study the geographic origin of the B.1.1.519 variant and its evolutionary relationship with the B.1.1.222 variant (Figure 3). The phylogenetic tree showed 3 defined clusters, two of which corresponded only to B.1.1.222 and B.1.1.519 variants, with clear separation, and a mixed cluster displaying a non-clearly defined separation among lineages. Thus, the detailed evolution of this SARS-CoV-2 lineage remains unclear. The mixed cluster is formed by the B.1.1.519 sequences most closely related to B.1.1.222 sequences. As part of the mixed cluster, we observed a clade formed by a small subclade of B.1.1.222 and a small subclade of B.1.1.519 sequences. Most B.1.1.519 subclade sequences were sequenced in the United States (4 out of 5) and one in Mexico City. Therefore, the geographic origin of the B.1.1.519 variant remains unclear.

#### 3.3.2. Haplotype Analysis

A haplotype network could provide new insights into evolutionary processes when external and internal nodes of a phylogeny are simultaneously studied. The continuous sequencing of SARS-CoV-2 samples throughout the pandemic enables the study of ancestral and child sequences simultaneously. A haplotype network for variant B.1.1.519 was constructed in this study to enable the analysis of how the evolutionary and mutational processes have impacted its dispersion and prevalence (Figure 4). A haplotype was defined based on all variable sites for the B.1.1.519 variant.

The prevalence of any haplotype was defined as the period of time (measured as the number of days) in which a haplotype was observed. The prevalence was defined as zero if a haplotype was observed in only one sample. The month of appearance corresponded to the month in which the first sequence of any specific haplotype was observed.

In Figure 4, the most ancient B.1.1.519 sequence can be observed as a blue-bordered node. This node can be used as an anchor to suggest the temporal direction of the haplotype network. In this representation, the size of a node is proportional to the prevalence of the haplotype. The most ancient sequence (blue-bordered large node, haplotype III) was observed in the largest number of samples (101 samples) during the longest period of time (190 days), which suggests a persistent and transmissible virus variant.

The results show that haplotype III diverged in the three next-largest nodes, which implies that these three haplotypes are the next-most commonly observed haplotypes (73, 64 and 56 samples, respectively). All of these secondary haplotypes showed persistent behavior across time (166, 187, and 139 days of prevalence, respectively). Interestingly, all of these haplotypes diverged into a large number of less efficient virus variants.

Additionally, the month of appearance was consistent with the peak in the effective reproductive number described earlier, as most haplotypes were first observed in November and some in December 2020.

### 3.4. Clinical Association

Finally, we studied the clinical impact of variant B.1.1.519. We analyzed the associations between variant B.1.1.519 and a number of clinical traits. Only sequences with complete clinical data were considered (*n* = 600). We found that patients infected with variant B.1.1.519 tended to show a significant increase in the odds of developing symptoms affecting the respiratory tract relative to non-B.1.1.519 variants. In particular, logistic regression models adjusted for covariates (age, sex, viral Ct and number of comorbidities) showed that variant B.1.1.519 was associated with a 1.786-fold increase in dyspnea (*p* = 0.0028, 0.202–0.964 95% CI), a 1.489-fold increase in chest pain (*p* = 0.035, 0.029–0.769 95% CI) and a 3.655-fold increase in cyanosis (*p* = 0.0456, 0.159–2.793 95% CI) (Table 1).

To investigate the relationship between variant B.1.1.519 and an increased risk of developing serious illness or death, we stratified patients into four age groups and compared their outcomes. Although we observed an overall trend of increasing disease seriousness with increasing age groups, infection with B.1.1.519 was still associated with a higher fraction of patients with serious illness and/or death than non-B.1.1.519 infection within each group (Figure 5).

We fitted logistic regression models to predict the severity of disease (see Section 2). After adjusting for covariates and various comorbidities, we still found that variant B.1.1.519 had a highly significant adjusted odds ratio (aOR) increase of 1.85-fold over non-B.1.1.519 variants (*p* = 0.000296, 1.33–2.6 95% CI) for developing a severe/critical outcome. Multivariate analyses adjusted for covariates also showed infections with variant B.1.1.519 to be associated with a 2.35-fold increase in the hospitalization rate (*p* = 0.005, 1.32–4.34 95% CI) (Table 2). Higher hospitalization rates and disease severity remained significantly associated with infection with the B.1.1.519 variant after removing asymptomatic patients (*n* = 20, 10 with the variant and 10 without) from the analysis (Appendix A). Among all symptomatic patients, dyspnea, cyanosis, chest pain, diarrhea and polypnea remained the most significant symptoms reported in B.1.1.519 vs. non-B.1.1.519, infections. 

## 4. Discussion

SARS-CoV-2 variant B.1.1.519 has been tagged by an alert for further monitoring by the WHO, implying that this variant could pose a future threat, but there is no evidence about phenotypic or clinical associations of concern. In this paper, we genomically described the B.1.1.519 variant and its evolution, transmissibility and clinical impact. The first patient carrying variant B.1.1.519 was detected in Mexico City in November 2020, representing the second case recorded worldwide. Three defined clusters were defined in the phylogenetic tree, two of them corresponding to B.1.1.222 and B.1.1.519 variants with a clear separation, and a mixed cluster. Finally, patients infected with variant B.1.1.519 seemed to show a significant increase in developing symptoms affecting the respiratory tract relative to those with non-B.1.1.519 variants. In addition, logistic regression models showed that variant B.1.1.519 was associated with an increase in dyspnea, chest pain, and cyanosis.

Worldwide, new variants of SARS-CoV-2 classified by the WHO as AFM have emerged. These variants, such as P.2 [30], B.1.621 [31], and B.1.1.318 [32], show spike mutations in receptor binding and S1/S2 cleavage sites and have spread widely within countries. By May 2021, B.1.1.519 had been detected in 31 countries and was predominantly found in Mexico (55%, 6041/10,922), followed by the USA (2.2%, 11,937/548,492), Canada (0.87%, 456/52,409), and Germany (0.14%, 192/130,634). B.1.1.519 has three substitutions in spike: T478K, P681H, and T732A. The S:T478K substitution is structurally localized in the region of interaction with the human ACE2 receptor. SARS-CoV-2 attaches to this receptor to infect cells, thus spreading the infection more effectively [33,34]. A study of in silico molecular dynamics on the spike has shown that the distribution of charges in S:T478K is most drastically affected at the site of substitution and its immediate vicinity on the surface of the folded protein. This effect may critically change the specific interactions with drugs, antibodies or the ACE2 receptor, increasing infectivity [35]. Accordingly, the Delta variant (B.1.617.2) carries the S:T478K substitution. This substitution could impact B.1.1.519 transmissibility and may help to explain why B.1.1.519 had a transmission advantage over other variants without S:T478K in Mexico City.

The S:P681H substitution is located immediately adjacent to amino acids 682–685, which correspond to a furin cleavage site at the S1/S2 binding site, where the more basic the string of amino acids is, the more effectively furin recognizes and cuts it [36], also this amino acid sequence serves as a cleavage site for the cellular host serine protease TMPRSS2, it plays an important role in promoting cell fusion, spread and pathogenesis in the infected host [34]. An in vitro assay with SARS-CoV-2 S:P681H using fluorogenic peptides mimicking the S1/S2 sequence reported an increase in spike cleavage by furin-like proteases but this does not significantly impact viral entry or membrane fusion [37]. This furin cleavage site is key to SARS-CoV-2 replication and pathogenesis because more furin cuts mean more spike proteins primed to enter human cells [38]. The Alpha (B.1.1.7) and Gamma (P.1) variants (recognized by the WHO as VOCs) carry S:P681H [37,39]. The S:P681H substitution could also be involved in the increased transmissibility of B.1.1.519 in Mexico City.

Genomic surveillance has proven to be an important tool for the identification and characterization of viral spreading potential and the monitoring of novel variants of concern in SARS-CoV-2 [16]. Based on genomic surveillance, we observed that variant B.1.1.519 showed increased transmission during the first and third weeks of December 2020 at the beginning of the second COVID-19 wave in Mexico City. After this increased transmission period, we estimate that B.1.1.519 became the dominant variant in circulation for the remaining period analyzed in this manuscript until late May 2021, completely displacing the previously dominant B.1.1.222. Such behavior is similar to that exhibited by other SARS-CoV-2 variants found in other regions [40].

Although we estimate two peaks of increased transmission for B.1.1.519, it could very well be that a low number of viable samples available for sequencing during epidemiological week 50 artificially split the transmission peak of B.1.1.519, given the behavior exhibited by other SARS-CoV-2 variants [41]. Regardless of this possible artifact, we estimate that the bulk of observed cases in Mexico City during the winter wave of COVID-19 were associated with variant B.1.1.519, with the associated clinical implications described in this manuscript.

Phylogenetic methods can be applied to provide some insight into the evolution and spread of SARS-CoV-2 [42]. However, conclusions drawn from phylogenetic and downstream analyses should be considered and interpreted with caution, as the sequences are too closely related. B.1.1.519 geographical origin could be inferred from the monophyletic group with its ancestor the B.1.1.222 lineage. However, the mixed inferred origin suggested fast dispersion due to human movement.

Evolutionary analysis of SARS-CoV-2 has been used to understand the spatiotemporal dynamics of the pandemic. Specifically, haplotype networks have been used to unravel the genetic diversity among monomorphic populations with small genetic distances between individuals, usually at the intraspecific level. Haplotype networks can be used to infer an evolutionary path for a given population. A median-joining network (MJN) is derived from a minimum spanning tree that traces a path between all studied sequences such that the total length is minimal. Additionally, an MJN will infer additional sequence types that minimize the inferred length; such inferred sequences can be considered biologically as unseen or extinct sequences. The distance between two sequence types will equal the number of nucleotide differences observed between them (Hamming distance) [43,44,45].

Recent studies have investigated the evolution and spatiotemporal distribution of SARS-CoV-2 via haplotype networks. Pereson, MJ, et al. studied the diversification of the spike protein in each SARS-CoV-2 clade, showing that two haplotypes predominated in specific clades (Hap-1 for clades G, GH and GR; and Hap-2 for clades L, O, S and V) [46]. In addition, sequence similarity and network structure were used to infer the import of SARS-CoV-2 from multiple countries in Bangladesh [47]. The edges in a haplotype network represent specific nucleotide substitutions, and the nodes represent specific sequence types or haplotypes. Garvin, MR, et al. inferred a haplotype network from 15,789 SARS-CoV-2 genomes to model their evolutionary success based on their duration, dispersal and frequency in the human population. They identified that the Pro323Leu mutation in the RNA-dependent RNA polymerase led to the rapid spread of the virus, rather than the previously reported Asp614Gly mutation in the spike glycoprotein. Importantly, they also inferred that the Pro323Leu mutation occurred on an Asp614Gly background [48].

The B.1.1.519 haplotype network shows a star-form, characteristic of an ongoing pandemic: ancestral central nodes surrounded by newly mutated peripheral nodes [49]. Continuous monitoring of SARS-CoV-2 genomes by this tool could highlight successful haplotypes with either high frequency, high prevalence or both. It could also highlight specific mutations responsible for increased transmission or prevalence. Indeed, some resources have been created to dynamically visualize haplotype networks of all worldwide SARS-CoV-2 genomes [50].

Finally, we observed that variant B.1.1.519 was significantly associated with severe disease, hospitalization, and death. This was particularly true with symptoms related to severe disease such as dyspnea, chest pain and cyanosis, which were more prevalent in B.1.1.519 compared with non-B.1.1.519 variant infections. We found these associations to be significant after correcting for the presence of common comorbidities such as diabetes, obesity and hypertension. Glycemic control status and ACE2 expression level have been previously associated with COVID-19 prognosis [5]. On the other hand, some anti-SARS-CoV-2 therapies, such as dexamethasone, have been associated with reduced mortality in hospitalized patients receiving respiratory support and Regeneron’s monoclonal antibody combination has been found to reduce deaths for hospitalized patients with severe COVID-19 who have not mounted their own immune response [51,52]. However, glycemic control status, ACE2 expression level and anti-SARS-CoV-2 therapy were not included in the clinical questionnaire. So, further studies would be required to study the role of these clinical and molecular characteristics in B.1.1.519 infection. Similarly, the Alpha VOC has been associated with an increased risk of hospitalization and greater disease severity or death [53,54]. Although there has been some contradictory evidence [55,56] concerning this point, more recent reports [50] have noted shortcomings of previous studies and reaffirmed the association of the variant with clinical severity. Some recent studies [57,58] have also shown that the Delta VOC is associated with an increased risk of hospitalization and severe illness/disease compared to infections with non-Delta variants that circulate at the same time. This increase, although smaller, is still significant compared to infections involving the Alpha, Beta and Gamma VOCs. Similarly, the Gamma VOC has also shown an increased risk of hospitalization [59] and severity in young adults with pre-existing conditions [60]. There is little evidence relating the Beta VOC to more severe disease or death, with only one study [61] comparing differences in the first and second waves in South Africa as a proxy for the Beta variant showing higher in-hospital mortality.

## 5. Conclusions

Sustained genomic surveillance plays a decisive role in identifying newly emerging SARS-CoV-2 variants and guiding the decisions of the public health care system in a country. Detailed evolutionary analysis is important to understand the origin and progression of newly evolving variants. Any significant clinical associations could be of interest in pandemic handling and containment.

## Figures and Tables

**Figure 1 viruses-13-02182-f001:**
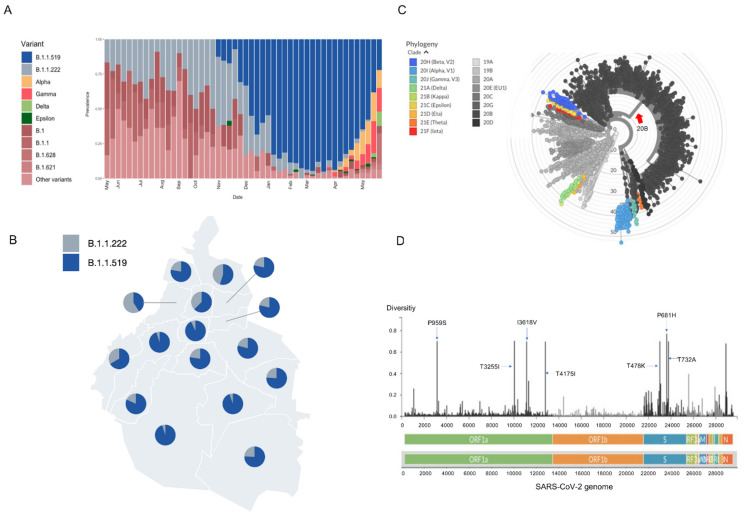
(**A**) Frequencies of the B.1.1.519 variant in Mexico City from May 2020 to May 2021. (**B**) The geographic distribution of B.1.1.519 and B.1.1.222 variants in Mexico City, with dominance of the first variant. (**C**) Phylogenetic tree of SARS-CoV-2 with NextClade clades. The branch indicated with a red arrow represents 1879 sequences of the B.1.1.519 variant of Mexico City with coverage of >99.5%. (**D**) Genome map of SARS-CoV-2 variant B.1.1.519 with the most representative amino acid substitutions in 1879 sequences of the B.1.1.519 variant of Mexico City with coverage of >99.5%.

**Figure 2 viruses-13-02182-f002:**
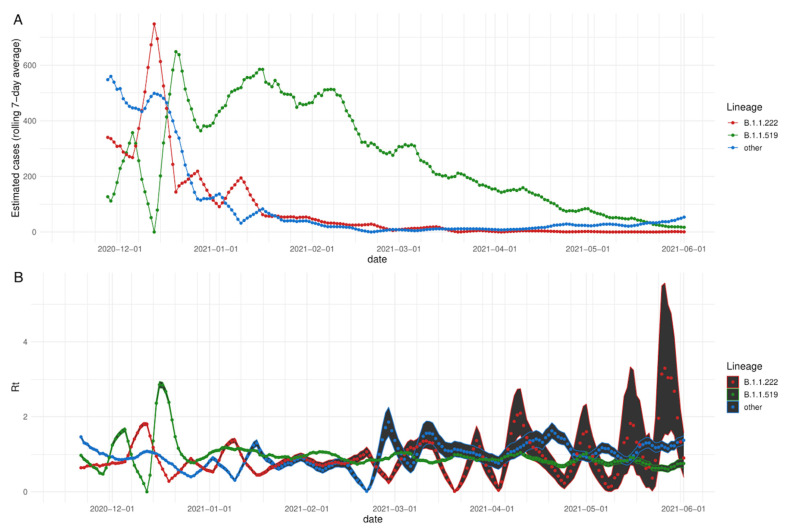
(**A**) Estimated number of cases for each variant based on the frequency observed in sequenced samples at INMG and the daily tally of confirmed cases in SINAVE, 7-day rolling average. (**B**) Time series of estimated Rt. Points represent the mean estimated Rt value per variant. Ribbon boundaries indicate the 5 (lower) and 95 (upper) quantile boundaries of the estimation.

**Figure 3 viruses-13-02182-f003:**
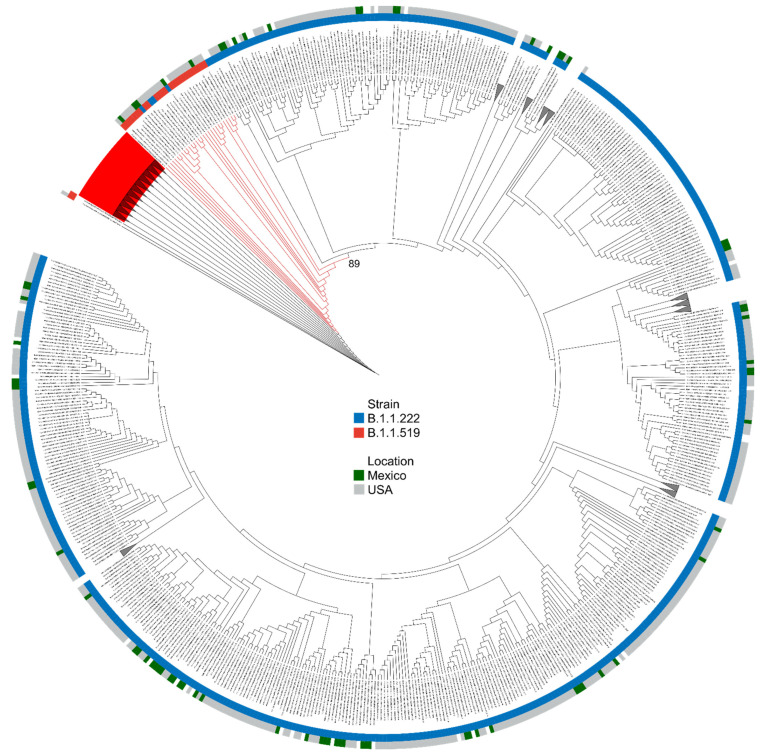
Phylogenetic relationships of SARS-CoV-2 B.1.1.222 and B.1.1.519 lineages. A maximum likelihood phylodynamic inference was done of 84 SARS-CoV-2 sequences from Mexico in a global background of 19312 sequences available in the GISAID EpiCoV database as of 1 May 2020. Leaves are colored according to their Pango lineage: B.1.1.519 (red) and B.1.1.222 (blue) and according to their geographical origin: Mexico (green) and USA (gray). The bootstrap value of the mixed cluster (described in the main text) is shown.

**Figure 4 viruses-13-02182-f004:**
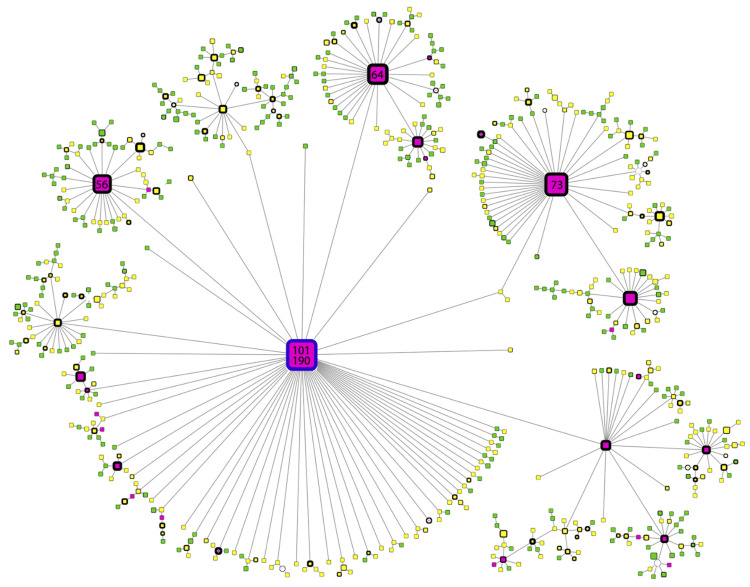
Haplotype network of B.1.1.519 sequences. Node colors represent the month of appearance (pink- fuchsia: November or December, yellow: January, February or March; green: April or May). Node size is proportional to the number of samples for that specific haplotype, and border width is proportional to the prevalence of the haplotype. Numbers correspond to the number of samples for that specific haplotype. The blue-bordered node indicates the haplotype with the most ancient appearance date for lineage B.1.1.519.

**Figure 5 viruses-13-02182-f005:**
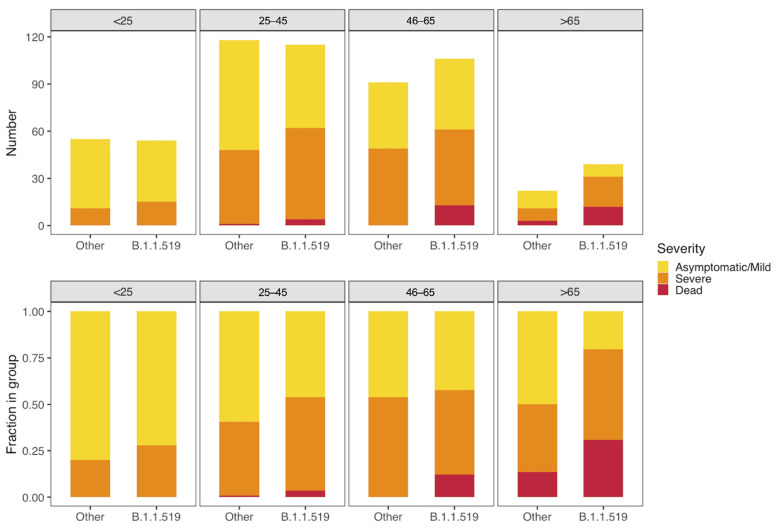
Severity of illness across patient age groups and by presence of B.1.1.519 or non-B.1.1.519 SARS-CoV-2 infections. The figure shows absolute counts (**upper**) and proportions of patients (**lower**).

**Table 1 viruses-13-02182-t001:** Associations between symptoms and variant B.1.1.519 using multivariate LR adjusted for covariates.

Symptom	B.1.1.519 *n* (%)	Other *n* (%)	aOR	95% CI	*p*-Value
Dyspnea	154 (49.0%)	103 (36.0%)	1.786	(0.202–0.964)	0.0028
Chest Pain	162 (51.6%)	117 (40.9%)	1.489	(0.029–0.769)	0.0350
Cyanosis	20 (6.4%)	9 (3.1%)	3.665	(0.159–2.793)	0.0456
Diarrhea	113 (36.0%)	91 (31.8%)	1.464	(−0.007–0.777)	0.0565
Polipnea	40 (12.7%)	46 (16.1%)	1.721	(−0.067–1.200)	0.0909
Myalgia	211 (67.2%)	181 (63.3%)	1.317	(−0.101–0.652)	0.1507
Rhinorrhea	98 (31.2%)	103 (36.0%)	0.757	(−0.661–0.106)	0.1549
Anosmia	173 (55.1%)	183 (64.0%)	0.769	(−0.636–0.107)	0.1647
Conjuntivitis	69 (22.0%)	89 (31.1%)	0.744	(−0.714–0.125)	0.1660
Odynophalgia	144 (45.9%)	143 (50.0%)	0.807	(−0.581–0.151)	0.2508
Arthralgia	196 (62.4%)	172 (60.1%)	1.235	(−0.159–0.581)	0.2628
Cough	204 (65.0%)	166 (58.0%)	1.233	(−0.167–0.584)	0.2740
Vomit	31 (9.9%)	26 (9.1%)	1.345	(−0.336–0.970)	0.3702
Persistant.Fever	47 (15.0%)	52 (18.2%)	0.800	(−0.719–0.279)	0.3780
Cephalea	213 (67.8%)	201 (70.3%)	0.861	(−0.553–0.249)	0.4651
Fever	185 (58.9%)	173 (60.5%)	0.925	(−0.456–0.298)	0.6868
Abdominal.Pain	31 (9.9%)	37 (12.9%)	1.011	(−0.591–0.636)	0.9714

**Table 2 viruses-13-02182-t002:** Association of the SARS-CoV-2 B.1.1.519 variant with disease severity and hospitalizations. The severity outcomes were coded as 0 = Asymptomatic/Mild, 1 = Severe, or 2 = Dead; an ordinary multivariate LR model was fitted adjusted for covariates. A binary multivariate LR model was fitted for hospitalization.

	Summary	Ordinal Multivariable LR Model (Severity)			Binary Multivariable LR Model (Hospitalization)		
Characteristic	*n* = 600 ^1^	aOR ^2^	95% CI ^2^	*p*-Value	aOR ^2^	95% CI ^2^	*p*-Value
Severity							
Asymptomatic/Mild	312 (52%)						
Severe	255 (42%)						
Dead	33 (5.5%)						
Hospitalized	69 (12%)						
Age	42 (29, 54)	1.04	1.03, 1.05	<0.001	1.06	1.04, 1.08	<0.001
Sex							
Female	302 (50%)	—	—		—	—	
Male	298 (50%)	1.21	0.87, 1.70	0.3	1.69	0.95, 3.04	0.075
Ct	19.13 (17.90, 20.40)	0.99	0.93, 1.06	0.8	1.05	0.95, 1.17	0.3
ImmunoSuppressed	18 (3.0%)	2.86	1.12, 7.42	0.029	2.43	0.59, 8.23	0.2
HD_Hypertension	107 (18%)	1.18	0.73, 1.90	0.5	1.55	0.82, 2.91	0.2
Diabetes	73 (12%)	0.91	0.53, 1.56	0.7	1.09	0.53, 2.14	0.8
Obesity	236 (39%)	1.42	1.01, 1.99	0.044	1.61	0.91, 2.87	0.10
Asthma	20 (3.3%)	1.53	0.62, 3.73	0.3	1.11	0.21, 4.43	0.9
Smoker	164 (27%)	1.21	0.83, 1.75	0.3	0.84	0.43, 1.57	0.6
Variant							
Other	286 (48%)	—	—		—	—	
B.1.1.519	314 (52%)	1.85	1.33, 2.60	<0.001	2.35	1.32, 4.34	0.005

^1^*n* (%); Median (IQR). ^2^ OR = Odds Ratio, CI = Confidence Interval.

## Data Availability

All genetic data is available at GISAID Initiative web page (https://www.gisaid.org/, accessed on 1 November 2020).

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
