# Peer review of "The Evolutionary Landscape of SARS-CoV-2 Variant B.1.1.519 and Its Clinical Impact in Mexico City"

_viruses, 2021, doi:10.3390/v13112182_

Round 1

Reviewer 1 Report

This study by Hidalgo-Miranda and Herrera et al. reports on the emergence of SARS-CoV-2 variants in Mexico City. The authors apply sequence data to assess geographical origin, reproduction, evolution, and clinical impact.

The methods utilized for viral detection, sequencing, and subsequent statistical interpretations appear to be properly employed.  The RT-qPCR primer/probe sets for 2019-nCoV_N1 and 2019-nCoV_N2 utilized for this study appear to have been well validated in independent reports.

The observed association between disease severity and variant B.1.1.519 appears to be well supported by the data presented and in line with other published reports.

Estimate number of cases for each variant are based on frequencies observed in sequenced samples against Federal databases. Samples were sequenced and the frequency of each variant determined, then the authors “extrapolated the total number of confirmed cases using the federal database for residents of Mexico City” The frequencies within the sample population are therefore critical to the findings reported here. More details on sampling would be helpful to determine if these 1835 samples are a suitable representation of the population. For example, were these samples obtained from patients admitted with COVID-19 symptoms or exposure (i.e. asymptomatic), were they located at one hospital, multiple hospitals (private or public), clinics, or a random sampling from a single location or numerous non-clinical sites throughout Mexico City?  While Figure 1B indicates 16 geographical locations throughout Mexico City, but more information on these sites (e.g. what they are and what type of patients they included) would be helpful.

The authors discuss the potential relevance of the P681H mutation on transmissibility, citing work by Lubinski et al demonstrating increased hydrolysis by Furin. It should be noted that this amino acid sequence also serves as a cleavage site for TMPRSS2, a membrane-bound host protease shown to also play an important role in promoting cell fusion, and discussed by Hoffmann et al. (29). More importantly, Lubinski et al demonstrated that, while the presence of this mutation increased proteolytic cleavage, no significant difference in viral entry or cell-cell spread was observed in furin or TMPRSS2-expressing functional models by the P681H mutation suggesting that other factors, including founder effects, may be responsible for its rapid progression.  

Overall, this study is well written and provides insight into variant frequencies in Mexico City during the time of sampling, their phylogenetic relation, and clinical impact.  

Reviewer 2 Report

-Please re-write the introduction, including a description of SARS-CoV2 pathogenesis, as example the intra-cellular entrance and replication via ACE2 and the activation of TMPRRS (Biomedicines. 2020 Oct 30;8(11):462. doi: 10.3390/biomedicines8110462). To date, this serin protein could be activated and regulated by epigenetic modulators (Biomedicines. 2020 Oct 30;8(11):462. doi: 10.3390/biomedicines8110462). please discuss this point.

-Please introduce data about the SARS-CoV2 infection and COVID19 in humans as overall, and as those with type 2 diabetes mellitus (T2DM); in this context the T2DM patients have worse prognosis (Diabetes Care. 2020 Jul;43(7):1408-1415. doi: 10.2337/dc20-0723; Cardiovasc Diabetol. 2020 Jun 11;19(1):76. doi: 10.1186/s12933-020-01047-y). This concept has been reported also for delta variants, and could be due to particular metabolic status of T2DM that works to enhance the ACE2 pathways in the glyceated form (Cardiovasc Diabetol. 2021 May 7;20(1):99. doi: 10.1186/s12933-021-01286-7). Please discuss this point.

-What was the pro-thrombotic status of patients. What was the percentage of intra-coronary acute thrombosis? This phenomena is valid in SAR-CoV2 patients and also in asymptomatic (ASAP) patients (Crit Care. 2021 Jun 24;25(1):217. doi: 10.1186/s13054-021-03643-0). Did you test the rate of SARS-CoV2 infected patients vs. ASAP patients in study cohorts?

-The study design has to be re-written, with more emphasis on the study poulation and the times of the study. 

-Again, what was the percentage of patients, and high risk patients treated with vaccination? Did you test the infection of variants in this study poopulation?

-Could you report the full medical anti-SARS-COV-2 therapies in the study population? Could it affect clinicla outocmes in study cohorts?

-If possible, I would see the full medical theapy in overall population, and in the cohorts as ASAP vs. non-ASAP, and in T2DM vs. non-T2DM, and hypertensive vs. non-hypertensive. Indeed, the worse glycemic control (Diabetologia. 2020 Nov;63(11):2486-2487. doi: 10.1007/s00125-020-05216-2), and the hypertension (J Am Heart Assoc. 2020 Sep;9(17):e016948. doi: 10.1161/JAHA.120.016948; BMC Cardiovasc Disord. 2020 Aug 14;20(1):373. doi: 10.1186/s12872-020-01658-z) could affect the the entity of SARS-CoV2 binding to human ACE2 receptor, and the thrombogenesis leading to worse prognosis. Please discuss it, and report the anti-hypertensive therapy in the study cohort.

Round 2

Reviewer 2 Report

Please improve English form of the text.